# Effects of Type and Content of Fibers, Water-to-Cement Ratio, and Cementitious Materials on the Shrinkage and Creep of Ultra-High Performance Concrete

**DOI:** 10.3390/polym14101956

**Published:** 2022-05-11

**Authors:** Ying Chen, Peng Liu, Fei Sha, Zhiwu Yu, Sasa He, Wen Xu, Maofeng Lv

**Affiliations:** 1School of Civil Engineering, Central South University of Forestry and Technology, 498 Shaoshan Road, Changsha 410004, China; cheny83@csu.edu.cn; 2National Engineering Laboratory for High Speed Railway Construction, Central South University, Changsha 410075, China; zhwyu0512@163.com; 3School of Civil Engineering, Central South University, 22 Shaoshan Road, Changsha 410075, China; 4Chenzhou Changxin Residence Technology Co., Ltd., 8 Huihuang Road, Chenzhou 423000, China; 5College of Engineering, Ocean University of China, 238 Songling Road, Qingdao 266100, China; 6Railway Group 5 Mechanization of Engineering Co., Ltd., 32 HongtangCong Road, Hengyang 421002, China; miaolx863@163.com (W.X.); l15116277646@163.com (M.L.); 7Prefabricated Construction Engineering and Technological Research Center of Hunan Province, 22 Shaoshan Road, Changsha 410075, China; 8Zhongda Design Institute Co., Ltd., 68 Shaoshan Road, Changsha 410075, China; csuhs@126.com

**Keywords:** ultra-high performance concrete, fiber types, shrinkage, creep

## Abstract

The effects of the type and content of fibers, water to cement ratio (*W/C*), and content of cementitious materials on the shrinkage and creep of ultra-high performance concrete (UHPC) were investigated. The relationships between curing age, shrinkage, and unit creep of the UHPC were also discussed. The results showed that the shrinkage of the UHPC decreased with the increase in *W/C*, where there existed a quadratic function between shrinkage and *W/C*. However, the unit creep of the UHPC increased with *W/C*. The shrinkage and unit creep of the UHPC increased with the increase in the content of the cementitious materials. The type and content of fibers had different effects on the shrinkage and unit creep of the UHPC, that is, the shrinkage of the UHPC first increased and then decreased with the increase in the content of steel fibers, where there existed a quadratic function between them. There was a linear function between the shrinkage of the UHPC and the content of carbon fibers, but the shrinkage of the UHCP first increased and then decreased with the increase in PVA content. The shrinkage and unit creep of the UHPC at the initial curing age were significant, which tended to be constant with the increase in curing age. Although the steel fibers had a significant inhibiting effect on the unit creep of the UHPC, the carbon fibers and PVA had positive and negative effects on the unit creep of the UHPC. The effects of the type and content of fibers on the shrinkage and unit creep of the UHPC were caused by the slenderness ratio, shape, surface roughness, and elasticity modulus of the fibers. The shrinkage and creep of the UHPC were caused by the chemical autogenous shrinkage and free water evaporation of the UHPC.

## 1. Introduction

Compared with normal concrete, ultra-high performance concrete (UHPC) is considered to be a new development of concrete technology with excellent workability, high compressive strength, toughness, long durability, etc. ASTM C1856 defines UHPC as a cementitious mixture with a specified compressive strength of at least 120 MPa [1], but GB/T 31387 considers that the compressive strength of UHPC is more than 100 MPa [2]. In general, UHPC made of superplasticizers, silica fume, and fibers has been widely used to construct super-high building structures, long-span bridges, and subsea tunnels [3]. Plenty of studies were carried out to discuss the preparation and performance of UHPC [4]. For example, Lankard et al. [5] made UHPC with a compressive strength of 238 MPa. Roux et al. [6] investigated the carbonation, chloride ion penetration, and wear-resisting properties of UHPC. The results showed that various performances of UHPC are superior to those of the normal concrete. Therefore, UHPC has both important academic meaning and engineering value.

Past studies focused on the mix proportion of concrete, material compositions, mechanical properties, preparation technology, and methods [7]. For example, Yu et al. [8] investigated the mix design and properties assessment of UHPC based on the densely compacted cementitious matrix and the modified Andreasen and Andersen particle packing model. Siwinski et al. [9] proposed a modification of Larrard’s formula for both the design and compressive strength evaluation of UHPC. The proposed modification consisted of the introduction of new parameters into the original formula that allowed it to consider the amount of binders and fine-grained aggregates, the amount of reinforcing fibers, the specimen shape and size, and the curing time, as well as provide a reinterpretation of the water/cement ratio. Carey et al. [10] evaluated the effects of constituent proportions on the mechanical and thermal properties of UHPC. However, few studies were carried out to investigate the long-term performance and volume stability [11].

The shrinkage and creep of concrete are the most commonly used parameters of the long-term performance of concrete. Usually, the reasons for concrete shrinkage are the chemical autogenous shrinkage caused by cement hydration, plastic shrinkage of water loss in a plastic state of paste, and drying shrinkage of hardened paste [12]. The concrete shrinkage can result in microcracks and damage, which affects the mechanical properties and various other performance factors of concrete [13]. Under sustained loads with a fixed value, the deformation of concrete will continue to increase as time increases; this phenomenon is called concrete creep. Currently, there are several theories to explain the phenomenon of concrete creep, such as viscoelasticity theory, seepage theory, viscous flow theory, plastic flow theory, micro-fractures theory, and internal forces balance theory [14]. In general, suitable shrinkage and creep of concrete can promote coordinated deformation of the structure, redistribution of the internal force, and reduction in the stress concentration, which has a positive effect on concrete structures. However, more excessive shrinkage and creep of concrete can bring negative effects, such as excessive deformation, concrete cracking, and loss of the prestress of concrete structures [15,16].

Concrete shrinkage and creep under variable hydric conditions are important for the safety and durability of concrete structures; therefore, plenty of studies were conducted. For example, Wang et al. [17] presented the results of brittle creep-relaxation experiments performed on concrete. Wu et al. [18] discussed the mechanism of autogenous shrinkage of high-performance concrete and influential factors in its development. Koh et al. [19] investigated the shrinkage properties of UHPC developing a compressive strength larger than 180 MPa, and the results showed that UHPC experienced very small amounts of drying shrinkage with extremely high autogenous shrinkage. The final values of shrinkage and creep of UHPC were influenced by numerous factors: temperature and humidity [20,21], cementitious materials [22], water-to-cement ratio (*W/C*) [23], fiber [24,25], granulometric and mineralogical composition of aggregates [26], shrinkage-reducing agent [19], concrete strength, pore structure [27], external load [28], method of workability and curing [29], concrete age at the end of curing, and many other factors [30]. Some important results were achieved by researchers, for example, Lam et al. [31] discussed the influence of basalt fiber content on some properties of UHPC, such as workability, compressive strength, flexural strength, elastic modulus, and shrinkage. The results showed that the flexural strength, compressive strength, and elastic modulus of the UHPC increased with the addition of the basalt fiber of reasonable content without affecting the workability of the concrete mixture. Garas et al. [32] discussed the effect of the stress/strength ratio, steel fiber reinforcement, and thermal treatment on the tensile creep and free shrinkage deformations of ultra-high-performance concrete. The results showed that the effects of both thermal treatment and fiber reinforcement were more pronounced in tensile creep behavior than tensile strength results of different UHPC mixes. Garas et al. [33] presented the characterization of the tensile creep of UHPC under different thermal treatment regimens, with a complimentary assessment of the underlying mechanisms via the characterization by nanoindentation and scanning electron microscopy. Moreover, some concrete creep models were proposed [34,35,36], for example, the ACI 209 model and MC 2010 [37,38]. Forth [39] developed a model that allowed for the prediction of tensile creep based on the compressive strength of the concrete, the applied stress level, and the relative humidity. Zhu et al. [40] developed the ABAQUS user subroutine for simulating the creep and shrinkage of both normal concrete and UHPC via the recursive algorithm of an adjacent stress increment in the time history.

Although UHPC has very high strength and excellent durability performance due to its dense microstructures, early-age cracks may occur due to the high heat of hydration and autogenous shrinkage caused by low *W/C* and high unit cement content [41]. The creep and shrinkage effect have a significant influence on the long-term behavior of UHPC, which limits the engineering application of UHPC. Moreover, amounts of cementitious materials were used to prepare UHPC with low *W/C*, which resulted in the difference in thermodynamics and kinetics between UHPC and normal concrete. So far, no consensus on the shrinkage and creep of UHPC was unified. The traditional UHPC was composed of quartz sand used as coarse aggregate, cement and admixtures regarded as cementitious materials, and water reducer, which demonstrated some disadvantages, including high cost and complex construction technology. In the absence of high-strength granite aggregate and silica sand, it is a challenge to prepare UHPC during on-site engineering. In previous research [42], we used the limestone gravel and river sand as coarse and fine aggregates, respectively, to prepare UHPC with a conventional preparation process, and the mechanical property of UHPC was also studied. In order to reveal the change law of long-term deformation performance, the effects of *W/C*, type and content of fibers, and dosage of cementitious materials on the shrinkage and creep of UHPC were investigated in this study.

The objective of this study was to indicate the relationship between compositions and long-term deformation performance of UHPC prepared with ordinary limestone gravel and river sand. This research established a theoretical basis and technical support for UHPC engineering applications. In this study, the shrinkage of UHPC was the sum of the chemical autogenous and drying shrinkages, and the deformation under test load minus the shrinkage was regarded as the creep of UHPC.

## 2. Experimental Procedure

### 2.1. Raw Materials

The P·O 52.5R early strength ordinary Portland cement was supplied by the Jiangxi Company and the I-level fly ash was produced by the Xiangtan Power Plant. The silica and organic silicon defoamer were purchased from Shanxi Company and Hengfa Company, respectively. The specific area and average size of silica were about 24.7 m^2^/g and 0.17 μm, respectively. The particle sizes of I-and Ⅱ-type limestone used as coarse aggregates were 5 mm~20 mm and 10 mm~30 mm, respectively. The carbon fibers were produced by Toho Carbon Fiber Co., Ltd. (Sandao, Japan) The polyvinyl alcohol fibers (PVA) and polypropylene fibers (PP) were obtained from Shanghai Chenqi Company and Hunan Huixiang Company, respectively. The hooked end steel fiber was provided by Hebei Demai Co., Ltd. (Shijiazhuang, China) The characteristic parameters of fibers are listed in Table 1. Moreover, the river sand with a fineness modulus of 2.3, tap water, and polycarboxylic acid series of high-efficiency superplasticizers containing a solid content of 30% were also used to prepare the UHPC. Table 2 shows the mix proportion design of UHPC.

### 2.2. Experimental Process and Testing Method

According to GB/T 50080-2016 [43] and GB/T 50081-2016 [44], the UHPC specimens with sizes of 100 mm × 100 mm × 100 mm, 100 mm × 100 mm × 400 mm, and 100 mm × 100 mm × 515 mm were cast to test the compressive strength, creep, and shrinkage. The specimens were cured at a standard condition for 24 h, i.e., a temperature of (20 ± 1) °C and RH of (95 ± 5)%. Then, the UHPC specimens were demolded and cured for different test ages at (20 ± 1) °C.

Shrinkage testing of the UHPC specimen was performed according to GB/T 50082-2009 [45]. The shrinkage of UHPC specimens of 100 mm × 100 mm × 515 mm was tested using the contact method. The cylindrical probes with a diameter of 6 mm were installed on both ends of the UHPC specimen. The initial length of the UHPC specimens was measured after being demolded. Then, the UHPC specimens were placed in a room environment for different ages, and the corresponding lengths were recorded. Each UHPC specimen was tested three times, and the average value was regarded as the length for different ages. There were three samples for each group. The shrinkage-testing process of UHPC is shown in Figure 1. The corresponding shrinkage of a UHPC specimen can be calculated using the following formula:(1)$$\varepsilon_{\mathrm{st}}=\frac{L_{0}-L_{t}}{L_{b}}$$
where $\varepsilon_{\mathrm{st}}$ is the shrinkage of the UHPC specimen at the *t*th day, *L*_0_ is the initial length of the UHPC specimen after being demolded, *L*_t_ is the length of the UHPC specimen cured for *t* days, and *L*_b_ is the gauge length.

Creep testing of a UHPC specimen was performed according to GB/T 50082-2009 [45]. The UHPC specimens with a size of 100 mm × 100 mm × 400 mm were used to carry out the creep testing when the specimens were cured for 28 d. Two UHPC specimens were put together vertically and were placed into a creep meter. The 30% axial compression strength of each UHPC specimen was applied using a hydraulic jack, which was set as the external load during the testing process.

Two measure points for each UHPC specimen were arranged on the opposite sides, where the gauge length was 200 mm. The deformation was recorded using a displacement sensor in a room environment. There were two samples for each group; the creep-testing process of UHPC is shown in Figure 1.

The lengths of the UHPC specimens at different ages, i.e., 1 d, 3 d, 7 d, 14 d, 28 d, 45 d, 56 d, and 90 d, were recorded. The creep degree of a UHPC specimen is defined as the deformation per unit load and length was obtained as follows:(2)$$C\;(t)=\frac{\varepsilon_{\mathrm{ct}}}{\sigma }=\left(\frac{\Delta L_{0}-\Delta L_{t}}{L_{c}}-\varepsilon_{\mathrm{st}}\right)/\sigma$$
where *C*(*t*) is the unit creep of UHPC specimens loaded for *t* days, $\varepsilon_{\mathrm{ct}}$ is the creep deformation of a UHPC specimen at the *t*th day, and σ is the stress of a UHPC specimen provided by an external load. $\Delta L_{0}$ and $\Delta L_{t}$ are the initial and testing lengths of UHPC specimen at the *t*th day, respectively. *L*_c_ is the gauge length of a UHPC specimen for testing creep, i.e., 200 mm.

The SBY-64B-type curing chamber for cement specimen was manufactured by Tianjin Huatong Test Instrument Factory. The WAW-DP type of Universal Tester and SHT4106G series of Electric Servo-hydraulic Material Test system were produced by Shanghai Sansi Co. Ltd. in China. The HSP-540 type concrete shrinkage dilatometer for a UHPC specimen was produced by Cangzhou Yixuan Test Instrument Co., Ltd. (Cangzhou, China) The XBJ500C automatic concrete creep meter was made by Tianjin Gangyuan Test Instrument Factory.

## 3. Results and Discussions

### 3.1. Effects of W/C on the Shrinkage and Creep of the UHPC

The macro-performances of UHPC are closely related to the micro-characteristics. The *W/C* can obviously affect the type and composition of hydration productions, porosity, and morphology of the microstructure. Therefore, it is very significant to investigate the effect of *W/C* on shrinkage and creep of UHPC.

#### 3.1.1. Effect of W/C on Shrinkage of the UHPC

The shrinkage of UHPC with different *W/C* values, i.e., 0.16, 0.18, 0.2, 0.22, and 0.24, was studied. Figure 2 shows the shrinkage curves of UHPC with respect to *W/C*.

As seen in Figure 2, the shrinkage of UHPC increased with the increase in *W/C*, which behaved according to the following quadratic function:(3)$$y_{s}=ax^{2}+bx+c$$
where *y*_s_ is the shrinkage of UHPC with different *W/C* values; *x* is the *W/C*; and *a*, *b*, and *c* are the fitted parameters.

For the same cured age, the shrinkage and change rate of UHPC became significantly higher with the decrease in *W/C*. The decreasing trend of shrinkage of UHPC was more remarkable with *W/C* = 0.16~0.2. When *W/C* changed from 0.16 to 0.18, the shrinkage of UHPC at 7 d, 28 d, and 56 d reduced by 25.2%, 15.1%, and 20.6%, respectively. Correspondingly, the values reduced by 21.1%, 28.4%, and 12% when *W/C* increased from 0.18 to 0.2. However, the corresponding percentage reductions of shrinkage of UHPC were 22.8%, 20.6%, and 30.7%, respectively. This may have been due to the reason that the hydration degree of UHPC is low when the *W/C* is low. This is due to the reason that the hydration degree of UHPC is incomplete under low *W/C*, and the hydration products, including hydrated calcium aluminate and hydrated calcium silicate, are not insufficient [45,46]. Therefore, the hardened paste has weak resistance to external deformation. Simultaneously, the hydration products have an amount of free water and interlayer water, which can be consumed during the hydration and drying process. Having plenty of water absorbed by the porous hardened paste can generate additional pressure during the evaporation process, which can result in a drying shrinkage phenomenon. Compared with the higher *W/C* of 0.24, the amount of reactive water of UHPC was lower. Low *W/C* and the incorporation of supplementary cementitious materials can remarkably affect the pore structure, relative humidity, self-stress, degree of hydration, and interface structure; hence, this increases the shrinkage of UHPC. Moreover, the early hardening of UHPC is not sufficient, which can provide substantial channels for water evaporation among the hydrated cement particles, which allows for the shrinkage of UHPC. The above conclusions accord well with [47], i.e., the shrinkage of a specimen with *W/C* of 0.2 was basically at the same level. In general, the shrinkage mechanism of concrete is caused by the change in relative humidity in concrete. The development of moisture in concrete can be divided into two stages. First, the initial shrinkage of concrete is chemical autogenous shrinkage of concrete, when the relative humidity in concrete at an initial stage is about 100%. Only a part of the volume shrinkage caused by chemical reduction is translated into the macroscopic shrinkage of concrete, which is related to the elastic modulus of concrete. The higher the elastic modulus of concrete, the lower the shrinkage of concrete. The drying shrinkage of UHPC plays a dominant role when the relative humidity in concrete decreases. The corresponding mechanism of shrinkage of UHPC is the surface tension of capillary water inside the concrete. Furthermore, the shrinkage of concrete generates a micro-cracking effect. That is, the relative humidity in concrete is heterogeneous, and the drying interface of concrete surface moves inward. The final result shows that the tensile force appears on the concrete’s surface as the pressure in concrete, which leads to the softening of tensile stress or induces local microcracks in the surface concrete layer. It can also be seen from Figure 2 that the shrinkage of UHPC increased with the increase of curing age. The shrinkage of UPHC at the initial stage was more significant, and it tended to be a constant when the curing age was more than 56 d. The shrinkage of UPHC significantly increased from 28 d to 56 d, which may have been because the testing period was summer, and therefore, the room environment had a high temperature and low relative humidity, and significant amounts of the free water in the specimens were evaporated. The above result shows that the shrinkage of UHPC happened at an early stage. In order to reveal the relationship between shrinkage and curing age, the shrinkage curves of UHPC with curing age were plotted, as shown in Figure 3.

Figure 3 indicates that the shrinkage of UHPC increased with curing age, and there existed a quadratic function relating them, i.e., Equation (3). The longer the curing age, the higher the hydration degree of cementitious materials of the UHPC and the larger the volume shrinkage generated by hydration products. Moreover, more free water in the UHPC was evaporated with increasing curing age. Therefore, the shrinkage of the UHPC became larger with the increase in curing age. The curing age and *W/C* displayed a coupling effect on the shrinkage of the UHPC. The shrinkage of the UHPC with a *W/C* of 0.16 cured for 3 days was about 260 × 10^−6^, but its value after curing for 56 days was 470 × 10^−6^. It can also be seen from Figure 3 that the shrinkage of the UHPC with the same *W/C* could be affected by *W/C*. For example, the difference in shrinkage between the UHPC with a *W/C* of 0.16 cured for 3 days and 56 days was about 210 × 10^−6^. However, its corresponding value with a *W/C* of 0.24 was about 150 × 10^−6^.

#### 3.1.2. Effect of W/C on the Creep of the UHPC

The *W/C* can also influence long-term performance and behaviors. In this part, the effect of *W/C* on the creep of the UHPC was investigated, and the creep curves of the UHPC with *W/C* are shown in Figure 4.

As seen in Figure 4, the creep of the UHPC increased with the increase in *W/C*. The higher the *W/C* was, the larger the creep of the UHPC became. Simultaneously, there existed a quadratic function between the creep of UHPC and curing age. Generally, the concrete creep can be considered as a delayed elastic deformation, which rapidly develops at the initial loading stage. The thin colloid layer of hardened concrete is irregular, with unstable bonding and contacts. The creep of UHPC is regarded as a change in the structure of a solid, which may be caused by the dislocation movement, combination, and reintegration of gel particles. Various solid materials may also move or migrate from a high-stress zone to a stress-free zone, and the bond failure of hardened cement slurry and aggregate with high stress is the main reason for concrete creep. Free water in UHPC paste with a low *W/C* is reacted and consumed, which results in less water evaporation. Therefore, the unit creep of UHPC is dominated by *W/C*. There was a little change in the unit creep of the UHPC with a lower *W/C* ranging from 0.16 to 0.2. The chemical autogenous shrinkage of UHPC increases with the increase of *W/C*, which is due to more water reacting with cementitious materials and more hydration products being generated. Simultaneously, plenty of free water in UHPC can be easily evaporated and transferred, which results in a significant change in unit creep. The unit creep of UHPC increased with the increase in curing age, which was due to the deformation of hardened paste. The unit creep of the UHPC first increased with curing age and then tended to be a constant. The *W/C* had a significant effect on the unit creep of the UHPC. In general, the higher the *W/C*, the larger the creep of UHPC. This may be because the hardened paste in UHPC is denser with a low *W/C* and it has more non-deformability to resist the creep of UHPC. However, the unit creep of UHPC decreases if the *W/C* is too higher under the same loading stress ratio. This is due to the reason that the relative strength development rate of UHPC with high *W/C* is more than that with low *W/C*; therefore, the unit creep of UHPC with high *W/C* is less at the same external load. Hardened UHPC can be regarded as a composite elastomeric cement gel skeleton filled with liquid. At the initial stage of the cement slurry bearing the load, there is no immediate elastic deformation of the solid, which is due to the water in the pore being partially subjected to the force. With the increase in loading time, the continuous solidification and hardening of cementite materials, such as cement slurries, consumes the free water; therefore, the water in the capillary and gel pores flows from a high-pressure part to low-pressure place. The final result is that the hardened paste undertakes a greater load and experiences larger elastic deformation. The smaller the pores of the hardened paste, the larger the additional capillary pressure, and the greater the shrinkage of UHPC. The micro-diffusion of water in capillary and gel pores can affect the deformation of the microstructure of UHPC. The pore stress effect and micro prestress relaxation also generates additional pressure, which can lead to shrinkage and creep. Moreover, the dry environment can also display a Pickett effect on the deformation of UHPC, which can increase the shrinkage of concrete specimens.

In order to reveal the effect of *W/C* on unit creep of UHPC, taking the unit creep of UHPC specimens cured for 90 d as an example, the relationship between *W/C* and unit creep of UHPC was investigated. Figure 5 shows the results of the unit creep of UHPC cured for 90 d with different *W/C* values.

Figure 5 shows that the *W/C* had little effect on the unit creep of the UHPC cured for 90 d when the *W/C* was no more than 0.2. This was due to the reason that there were lower chemical autogenous and drying shrinkages of the UHPC when the *W/C* was too small. The unit creep of UHPC became larger when the *W/C* was larger than 0.2. It can also be seen in Figure 5 that the unit creep of UHPC cured for 90 d was small, and the corresponding values ranged from 9.2 × 10^−6^/MPa to 13.4 × 10^−6^/MPa. The above result showed that the UHPC had a wonderful creep resistance performance. The hardened UHPC can be composed of cement paste and inert aggregate. The cement paste can crawl and flow under the external load, but the inert aggregate cannot crawl. Therefore, the inert aggregate bears more load than hardened cement paste, which can prevent the slurry of cement paste in UHPC. The external load borne by hardened cement paste will be taken by inert aggregate when the loading time increases. Therefore, the unit creep of UHPC gradually tends to be a constant.

### 3.2. Effects of Content of Cementitious Materials on the Shrinkage and Creep of the UHPC

Plenty of cementitious materials, including cement, fly ash, and silica, have been used to prepare UHPC, which results in the difference in microstructure and macro-performance of UHPC. In order to reveal the relationship between the content of cementitious materials and deformation, the effects of cementitious materials on shrinkage and creep of UHPC were investigated.

#### 3.2.1. Effect of Cementitious Materials on the Shrinkage of the UHPC

The cementitious materials of different contents, i.e., 550 kg/m^3^, 600 kg/m^3^, 650 kg/m^3^, 700 kg/m^3^, and 750 kg/m^3^, were used to prepare the UHPC, and the corresponding shrinkages of UHPC cured for different ages were tested. Figure 6 shows the shrinkage curves of the UHPC with different contents of cementitious materials.

It can be seen from Figure 6 that the shrinkage of the UHPC increased with the increase in the content of cementitious materials. When the content of cementitious materials was no more than 650 kg/m^3^, the content of cementitious materials had little effect on the shrinkage of UHPC with the same curing age. However, the shrinkage of UHPC became significant when the content of cementitious materials was more than 650 kg/m^3^. The reason was that the chemical autogenous and drying shrinkages of UHPC became larger with the increase in the content of cementitious materials. Compared with the shrinkage curves of UHPC, a conclusion can be drawn that the content of cementitious materials ranging from 650 kg/m^3^ to 700 kg/m^3^ had a significant effect on the shrinkage of the UHPC. The cementitious minerals hydrate with water, and the silica and fly ash can be activated by calcium hydroxide (CH) to generate hydration products, which consumes the free water and results in chemical autogenous shrinkage of UHPC. Moreover, the drying shrinkage of UHPC appears under the action of an external dry environment. When the content of cementitious materials was more than 700 kg/m^3^, the shrinkage of the UHPC decreased. This was due to the reason that some cementitious materials cannot be hydrated and play a micro-aggregate filling effect in UHPC. It can also be seen from Figure 6 that the shrinkage of UHPC changed with curing age. In order to reveal the relationship between them, the shrinkage of UHPC prepared using different cementitious materials contents (CMCs) with curing age was investigated, as shown in Figure 7.

Figure 7 indicates that the shrinkage of the UHPC increased with the increase in curing age, and it changed significantly at the initial curing age of 10 d. The increasing rate of shrinkage of UHPC decreased when the curing age was more than 10 d. Finally, it tended to be a constant when the curing age was more than 40 d. The reason is that the shrinkage of UHPC is caused by the chemical autogenous and drying shrinkages of UHPC. More hydration products are generated by the self-hydration reaction of UHPC; therefore, the large chemical autogenous shrinkage of UHPC is generated. Moreover, plenty of free water in UHPC is evaporated at the initial stage. Therefore, the change rate of shrinkage of UHPC is larger. The hydration reaction and evaporation decreased with the increase in curing age; therefore, the corresponding shrinkage of UHPC first decreased and tended to be a constant. A conclusion can be drawn from Figure 6 and Figure 7 that the curing age and cementitious materials content (CMC) had significant effects on the shrinkage of the UHPC, and an appropriate content of cementitious materials can significantly control the shrinkage of the UHPC.

#### 3.2.2. Effect of Cementitious Materials on the Creep of the UHPC

The effect of the content of cementitious materials on the creep of the UHPC was investigated, where Figure 8 shows the unit creep curves of the UHPC with different contents of cementitious materials.

As seen in Figure 8, the unit creep of the UHPC approximately increased with the increase in the content of cementitious materials. Larger chemical autogenous and drying shrinkages of UHPC were generated when the content of cementitious materials was higher. The hardened paste has a weak resistance to deformation. Simultaneously, the coarse aggregate, which is considered a key component of UHPC, has a small creep and bears the main action of stress. The higher the content of cementitious materials, the lower the percentage of coarse aggregate in UHPC. Therefore, the unit creep of the UHPC increases with the content of the cementitious materials. It can also be seen from Figure 8 that the unit creep of UHPC increased with curing age and content of cementitious materials. This was because large chemical autogenous and drying shrinkages of UHPC occurred with increasing curing age. In order to discuss the relationship between unit creep and the content of cementitious materials, taking the unit creep of UHPC cured for 90 d as an example, the unit creep of the UHPC with respect to the content of cementitious materials was investigated, as shown in Figure 9.

As seen from Figure 9, the unit creep of the UHPC cured for 90 d first decreased and then increased with the increase in the cementitious materials content. The values of unit creep of the UHPC ranged from 9.77 × 10^−6^/MPa to 10.99 × 10^−6^/MPa when the content of cementitious materials was no more than 650 kg/m^3^. The corresponding unit creep of the UHPC cured for 90 d was about 13.13 × 10^−6^/MPa when the content of cementitious materials was no more than 750 kg/m^3^. The reason was that more cementitious materials reacted to produce hydration products, which resulted in the larger shrinkage. Figure 9 also indicates that the unit creep of the UHPC cured for 90 d was the least when the content of cementitious materials was 600 kg/m^3^, which implied that a suitable content of cementitious materials had a positive effect on the creep of the UHPC.

### 3.3. Type and Content of Fibers on the Shrinkage and Creep of the UHPC

Fibers can improve the performance in terms of tensile strength, flexural strength, and fatigue, and enhance the ductility and toughness of hardened paste. Moreover, a suitable content of fiber can also effectively prevent the expansion of microcracks in concrete and the formation of macrocracks. The study of the effects of fibers on the shrinkage and creep properties of UHPC can provide theoretical guidance and technical support for the application of UHPC.

#### 3.3.1. Effect of Fibers on the Shrinkage of the UHPC

The effects of the type and content of four fibers, namely, steel fibers, PP, PVA, and carbon fibers, on the shrinkage of the UHPC were investigated, and Figure 10 shows the shrinkage curves of UHPC with different types and content of fibers.

Figure 10 shows that the shrinkage of the UHPC first increased and then decreased with the increase of the content of steel fibers, and there existed a quadratic function that described the relationship between them. The shrinkage of the UHPC containing 1% steel fibers was more than that of the blank sample. However, its value was less than that of the blank sample when the content of steel fibers was more than 1%. The above results indicated that a suitable content of steel fiber played a positive effect on the shrinkage of the UHPC. The elasticity modulus and tensile strength of steel fiber are high, which can effectively improve the connection, deformation, and crack-resistance performances of hardened paste. Steel fibers were not able to improve the microstructure and form a network when the content of steel fiber was no more than 1%. Therefore, the crack resistance effect of the steel fiber in the UHPC was not remarkable. The improving effect of steel fiber on the shrinkage of the UHPC became more significant when the content of steel fiber was more than 1%. As seen from Figure 10b, the shrinkage of UHPC first decreased and then increases with the increase in PP content when the curing age was less than 28 d. However, the shrinkage of the UHPC cured for 56 d first increased and then decreased, and there existed a quadratic function that described the relationship between them. Figure 10c,d shows that the shrinkage of UHPC was a positive linear relationship with the content of carbon fiber. However, the corresponding shrinkage of the UHPC first increased and then decreased with the content of PVA, and a quadratic function could be used to describe their relationship. The types and contents of fibers had different effects on the shrinkage of the UHPC, which were caused by the difference in slenderness ratio, interfacial bond strength, and elasticity modulus of fibers. The fibers in UHPC have less draw force and restraining action to restrain the larger deformation, which cannot form a network structure when the fiber content is low. A suitable content of fibers in UHPC can enhance the connection of hardened paste and improve the resistance to cracking and shrinkage. However, too much fiber content causes dispersion and agglomeration problems, which cannot reduce the shrinkage of UHPC. Compared with the shrinkage curves of four fibers in Figure 10, a conclusion can be drawn that the PP with a content of 2‰ had the best inhibitory effect on the shrinkage of the UHPC. Figure 10 also shows that the shrinkage of UHPC increased with the increase in curing age. This was due to the reason that the chemical autogenous and drying shrinkages of the UHPC increased with curing age.

In order to reveal the relationship between shrinkage of UHPC and fibers, the effects of the types and contents of fibers on shrinkage of UHPC were investigated. Figure 11 shows the shrinkage curves of the UHPC with different fibers and curing ages.

Figure 11 shows that the shrinkage of the UHPC increased with the increase in curing age and then tended to be a constant. The longer the curing age, the larger the shrinkage of the UHPC that was generated by the chemical autogenous and drying shrinkages. The change rate of shrinkage of the UHPC is significant at an initial curing age of 21 d, and it tended to be a constant when the curing age was more than 28 d. The change rule of shrinkage of UHPC with different fibers was basically similar, and the difference was shown as the difference in the final shrinkage. The steel fibers could play an inhibiting effect on the shrinkage of the UHPC, but the other three fibers could increase the shrinkage of the UHPC. Among them, carbon fiber has the most significant effect on the increase in the shrinkage of the UHPC. As seen from Figure 11b, the shrinkage of the UHPC with PP was less than that of the blank sample when the initial curing stage was no more than 40 d, but it became the opposite when the curing age was more than 40 d. The above results implied that PP had a positive effect on restraining the shrinkage of the UHPC at an initial curing stage. The shrinkage of the UHPC was larger than that of the blank sample with the increase in curing age. It was because the PP elasticity modulus was less than 3.5 GPa, which could only restrain the shrinkage of the UHPC during the initial curing stage. The chemical autogenous and drying shrinkages became larger at the later stage; therefore, the small elasticity modulus of PP could not effectively restrain the deformation of UHPC. Figure 11d indicates that the shrinkage of the UHPC increased with the increase in the PVA content when the content of PVA in UHPC ranged from 1‰ to 3‰. The shrinkage of the UHPC with and without PVA was not remarkable at the initial curing age. However, the shrinkage of the UHPC first increased and then decreased with the increased content of PVA. The greatest shrinkage of UHPC appeared when the content of PVA was 1‰. The greater the content of fibers, the more difficult the dispersion of fibers. Therefore, it was easier to cause agglomeration, which weakened the restraining action of fibers on the shrinkage of UHPC. To sum up, the types and contents of fibers had different effects on the shrinkage of the UHPC. Therefore, the shrinkage of UHPC can be controlled by adjusting the fibers.

In order to reveal the effect of curing age on the shrinkage of the UHPC, the shrinkage of the UHPC cured for 7 d and 56 d was investigated. Figure 12 shows the shrinkage curves of UHPC cured for 7 d and 56 d.

As seen in Figure 12, the steel fibers and PP had an inhibiting effect on the shrinkage of the UHPC cured for 7 d, and both of them had the same effect on the shrinkage of the UHPC. The shrinkage of the UHPC was in a range of 120~140 × 10^−6^, with the content of steel fibers and PP of 2%~3% and 1‰~3‰, respectively, which were 23.5%~34.4% smaller than the blank sample. The shrinkages of the UHPC with 2% and 3% steel fibers were 9.1% and 7% relative to the blank sample, respectively, when the UHPC was cured for 56 d. The carbon fibers had a low inhibitory effect on the shrinkage of the UHPC. The shrinkage of the UHPC cured for 56 d with 3‰ carbon fibers was about 491 × 10^−6^, which was more than 65% of the blank sample.

#### 3.3.2. Effect of Fibers on the Creep of the UHPC

The type and content of fibers can also affect the creep of UHPC. The unit creep of the UHPC with different content of fibers and curing age was investigated, and the corresponding results are shown in Figure 13.

Figure 13 indicates that the steel fibers had a significant inhibiting effect on the creep of the UHPC, but the PP could enhance the creep of the UHPC. The carbon fibers and PVA had double effects on the creep of the UHPC, i.e., positive and negative effects, which were due to the contents of both fibers. The creep of the UHPC gradually decreased when the content of the steel fibers was no more than 2%. However, the unit creep of the UHPC increased with the increase in the content of steel fibers. To sum up, the steel fibers had an inhibiting effect on the creep of the UHPC. The unit creep of the UHPC first decreased and then increased with the increase in carbon fiber content. The carbon fibers had a wonderful inhibiting effect on the unit creep of the UHPC when the volumetric ratio was 1‰. However, the carbon fiber caused an increase in the creep of the UHPC when the content of carbon fiber increased to 3‰. The content of PVA had little effect on the unit creep of the UHPC, and it enhanced the unit creep of the UHPC with a content of 3‰. However, the content of PP had a remarkable effect on the unit creep of the UHPC, and the change rate of the unit creep of the UHPC became more significant with the increase in PP. This may have been due to the difference in the characteristics of the fibers, such as the slenderness ratio, roughness, shape, and elasticity modulus. Moreover, adding more fibers to fresh concrete can easily cause an agglomeration phenomenon, which weakens the restraining action of fibers on the UHPC creep.

In order to reveal the relationship between the fibers and the creep of the UHPC, taking the UHPC cured for 90 d as an example, the effects of the type and content of fibers on the unit creep of the UHPC were investigated. Figure 14 shows the unit creep curves of the UHPC cured for 90 d with the increase in fiber contents.

Figure 14 shows that the types and contents of fibers had different effects on the unit creep of the UHPC. The unit creep of the UHPC first decreased and then increased with the increase in steel fiber, and its value was less than that of the blank sample. The unit creeps of the UHPC with carbon fibers and PVA were less than that of the blank sample when the content was no more than 2‰, and there existed the minimum unit creep of UHPC with a content of 1‰. The above results implied that a low addition of carbon fibers and PVA in the UHPC had an inhibitory effect on the unit creep. However, the unit creep of the UHPC increased with the increase in PP, and it was more than that of the blank sample. The above phenomenon was caused by the characteristic of fibers, such as elasticity modulus, surface roughness, shape, and slenderness ratio. Moreover, the bonding strength and contacting conditions between fibers and hardened paste were the other reasons. The hardened paste of the UHPC easily generated creep and shrinkage under the effect of loading. This was due to the reason that the low elasticity modulus of PP is less than 3.5 GPa, and thus, it cannot effectively restrain the creep of UHPC. A conclusion can be drawn that the fiber can restrain the creep when its elasticity modulus is larger than that of the matrix. Conversely, adding fiber increases the creep of UHPC. The above conclusion agrees well with [48]. Moreover, adding plenty of PP to UHPC results in the problem of reunion and dispersion, and a large number of initial defects are formed in UHPC. In general, the addition of fibers in UHPC can lead to energy consumption by deformation and bond slipping by transferring the force, which can result in the stress redistribution of the microstructure and restrain the creep of UHPC. However, the unit creep of UHPC increases if the fibers’ negative effect dominates. Compared with steel fiber, carbon fiber and PVA are flexible fibers, which are hard to be dispersed. Therefore, they have a worse inhibiting effect on unit creep of UHPC.

## 4. Conclusions

Based on the results presented in this study, the following conclusions could be drawn:The *W/C* had a significant effect on the shrinkage and unit creep of the UHPC prepared using limestone gravel and river sand with a conventional preparation process. The shrinkage of the UHPC decreased with the increase in *W/C*, but the unit creep of the UHPC increased. There existed a quadratic function relating the shrinkage and *W/C*, as well as the unit creep and *W/C*. The change rate of shrinkage and unit creep of the UHPC at the initial curing age was significant, and it tended to be a constant with the increase in curing age. The smaller the *W/C* of the UHPC cured for the same age, the more significant the shrinkage and change rate of the UHPC. The shrinkage of the UHPC was caused by the chemical autogenous and drying shrinkages of hardened paste, and the creep of the UHPC was due to the slippage of hydration products and the deformation of the microstructure under the external load.The shrinkage and unit creep of the UHPC increased with the increase in the content of the cementitious materials. The content of cementitious materials had a significant effect on the shrinkage of the UHPC when the content of cementitious materials was more than 650 kg/m^3^. Plenty of hydration products are generated during the hydrating process, resulting in a larger chemical autogenous shrinkage when the content of cementitious materials is high. Moreover, the shrinkage of the UHPC increased with the increase in evaporated free water. The minimum unit creep of the UHPC cured for 90 d was found when the content of cementitious materials is less than 650 kg/m^3^.The types and contents of fibers had different effects on the shrinkage and unit creep of the UHPC. The shrinkage of the UHPC first increased and then decreased with the increase in the content of steel fiber, and there existed a quadratic function that described the relationship between them. The shrinkage of the UHPC was more than that of the blank sample when the content of steel fiber was 1%. There existed a linear functional relationship between the shrinkage of the UHPC and the content of carbon fibers. However, the shrinkage of the UHCP first increased and then decreased with the increase in the content of PVA. Moreover, the shrinkage of UHPC increased with the increase of curing age and then tended to be a constant. The steel fiber had a significant inhibiting effect on the unit creep of the UHPC. Adding PP into the UHPC increased the unit creep. The carbon fiber and PVA had positive and negative effects on the unit creep of the UHPC. The effects of the type and content of fibers on the shrinkage and unit creep of the UHPC were caused by the slenderness ratio, shape, surface roughness, and elasticity modulus of the fibers.

## Figures and Tables

**Figure 1 polymers-14-01956-f001:**
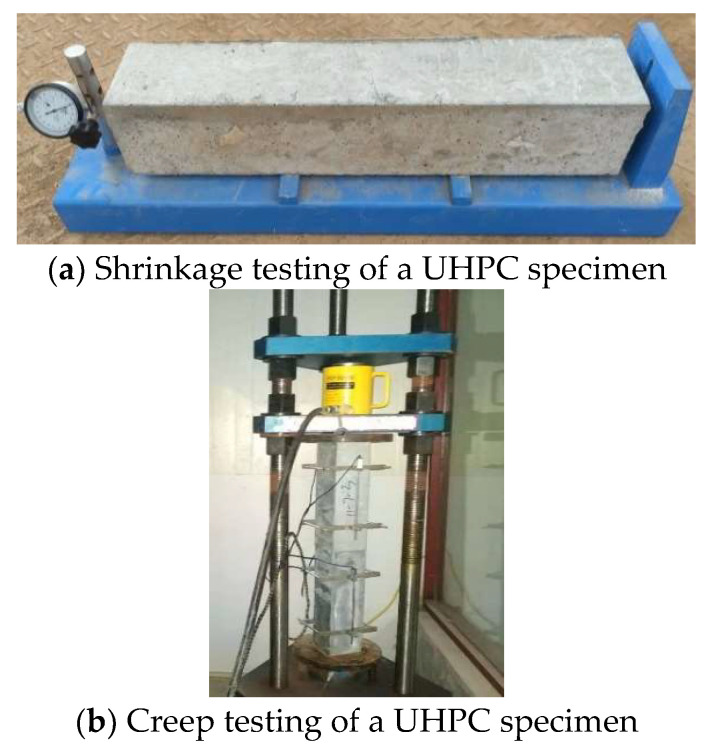
Shrinkage and creep-testing process of UHPC specimens.

**Figure 2 polymers-14-01956-f002:**
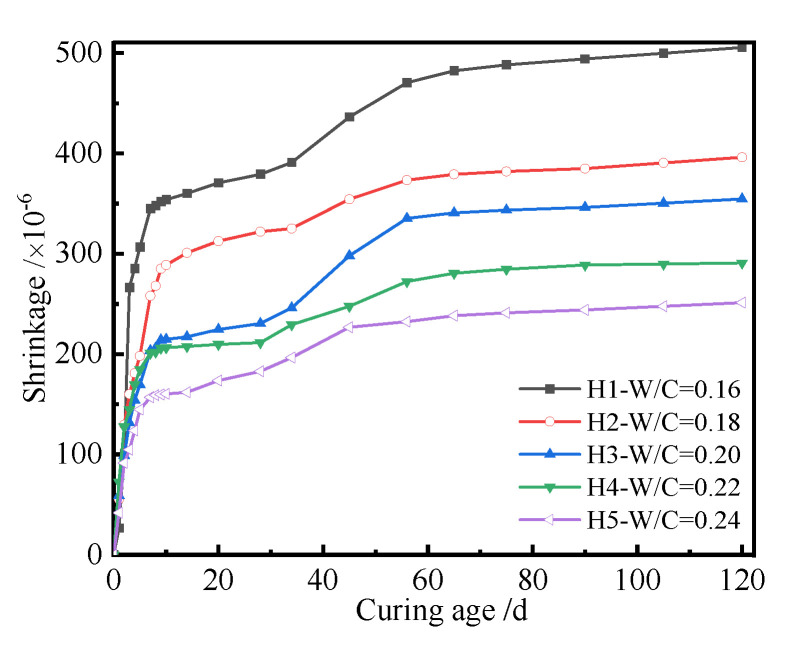
Shrinkage curves of the UHPC with *W/C*.

**Figure 3 polymers-14-01956-f003:**
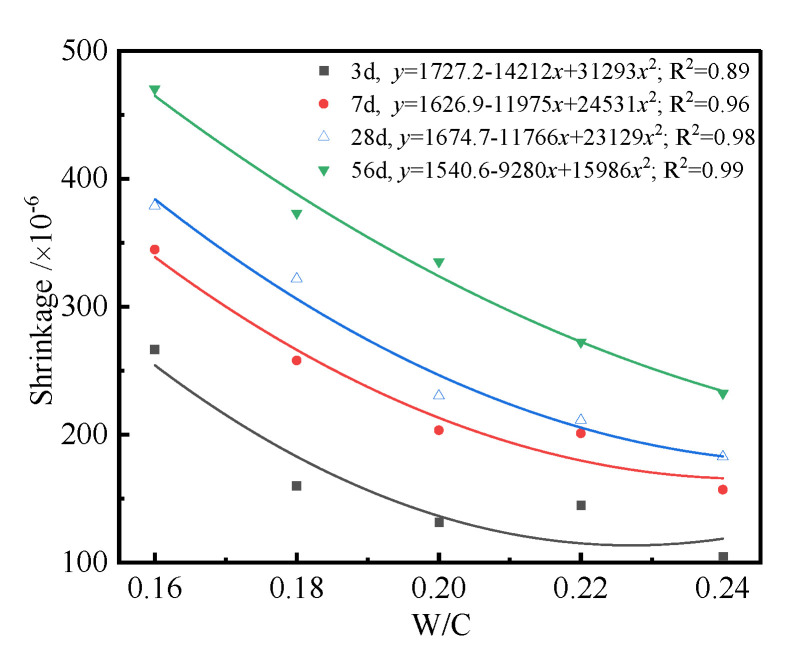
The shrinkage curves of the UHPC with curing ages.

**Figure 4 polymers-14-01956-f004:**
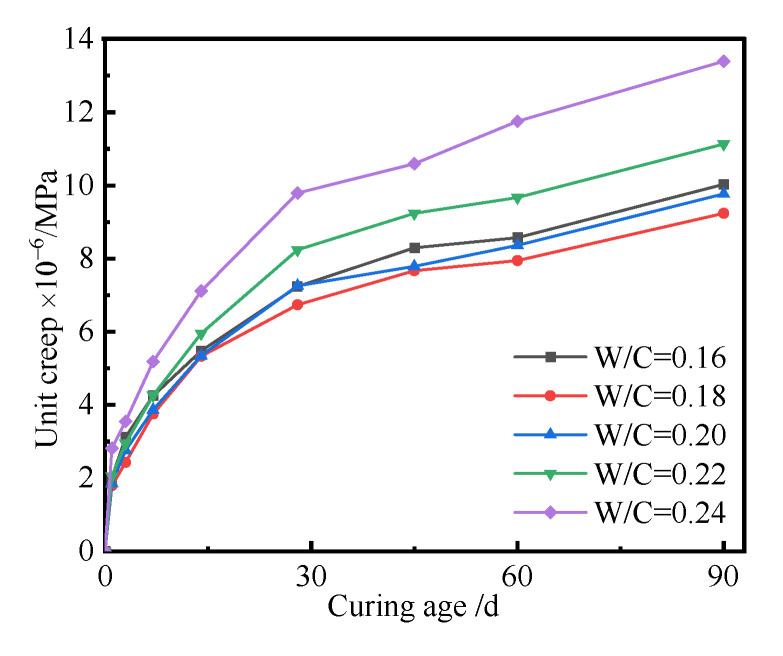
Creep curves of the UHPC with different *W/C* values.

**Figure 5 polymers-14-01956-f005:**
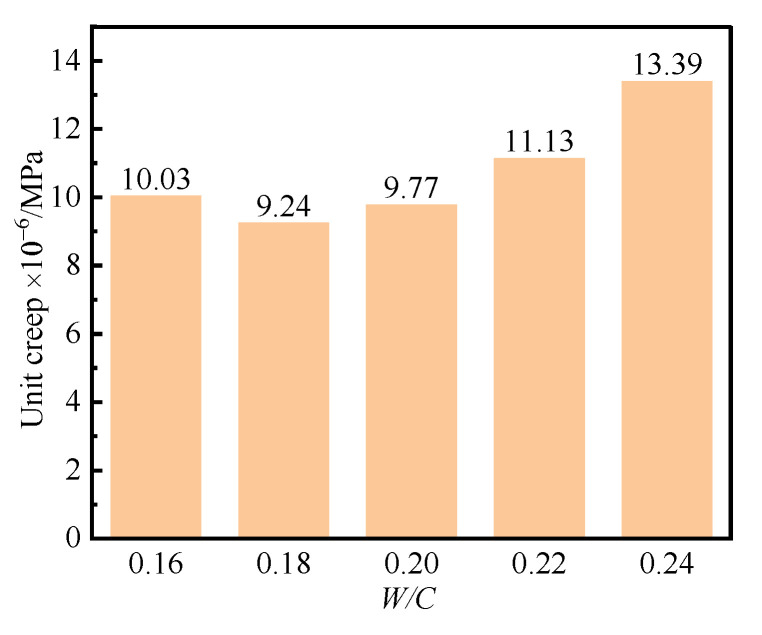
Unit creep of the UHPC cured for 90 d with different *W/C* values.

**Figure 6 polymers-14-01956-f006:**
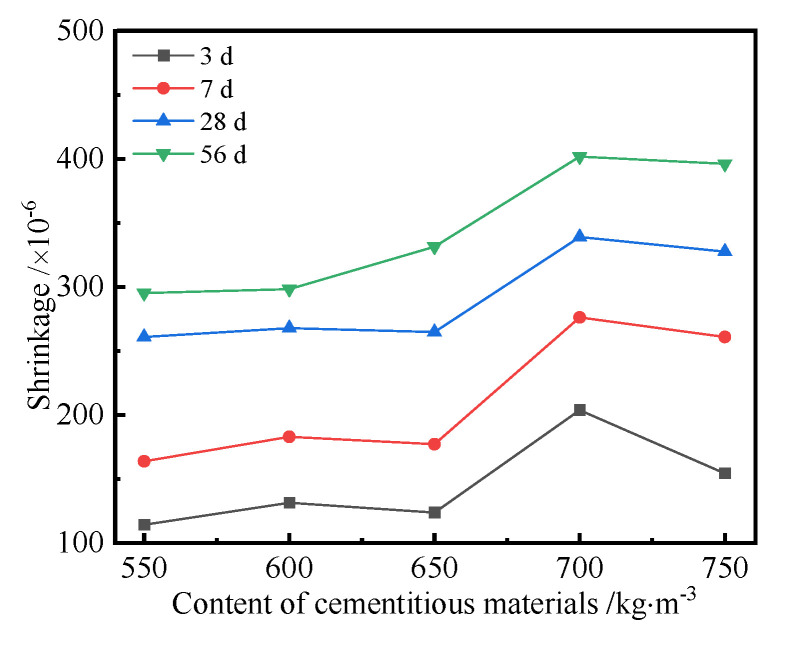
Shrinkage curves of the UHPC with different contents of cementitious materials.

**Figure 7 polymers-14-01956-f007:**
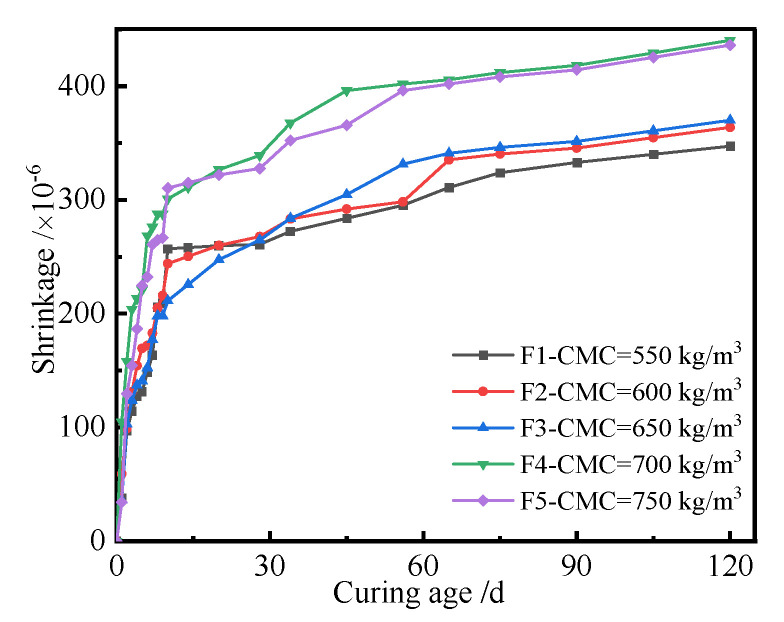
Shrinkage curves of the UHPC with curing age.

**Figure 8 polymers-14-01956-f008:**
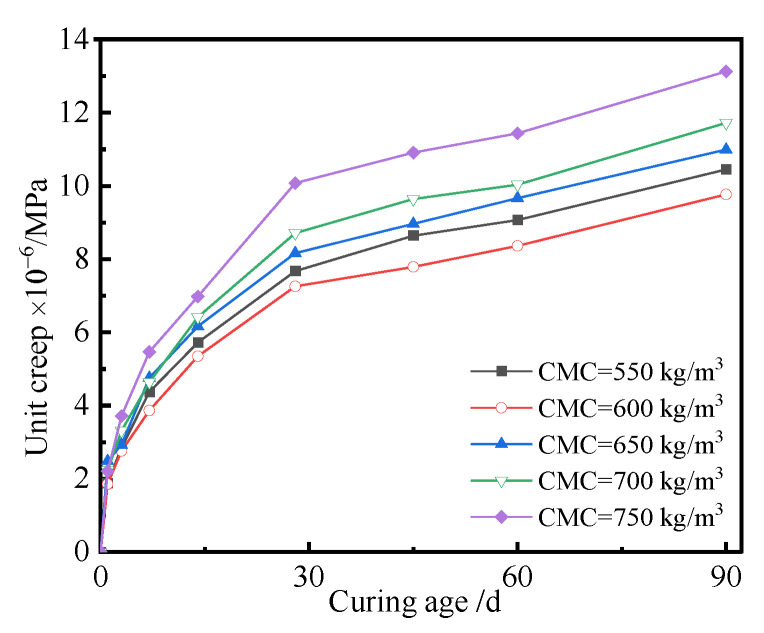
Unit creep curves of the UHPC with different contents of cementitious materials.

**Figure 9 polymers-14-01956-f009:**
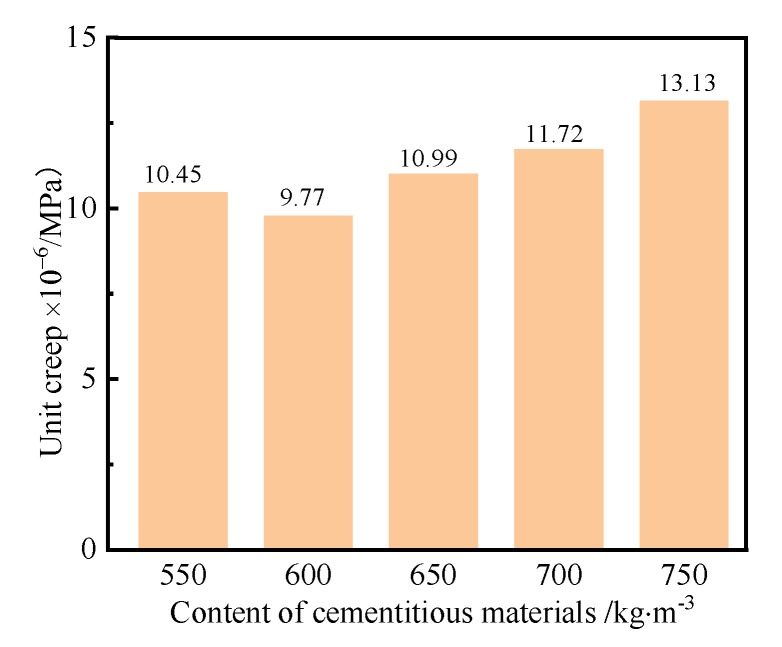
Unit creep of the UHPC cured for 90 d with different contents of cementitious materials.

**Figure 10 polymers-14-01956-f010:**
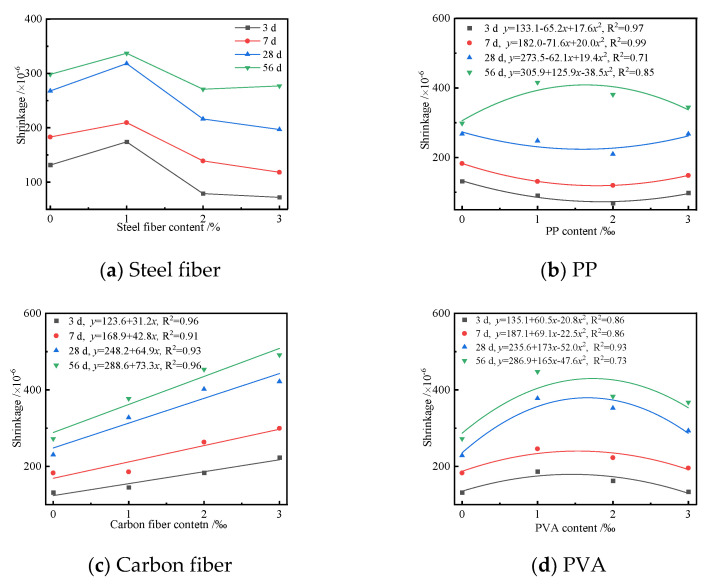
Shrinkage curves of the UHPC with different types and contents of fibers.

**Figure 11 polymers-14-01956-f011:**
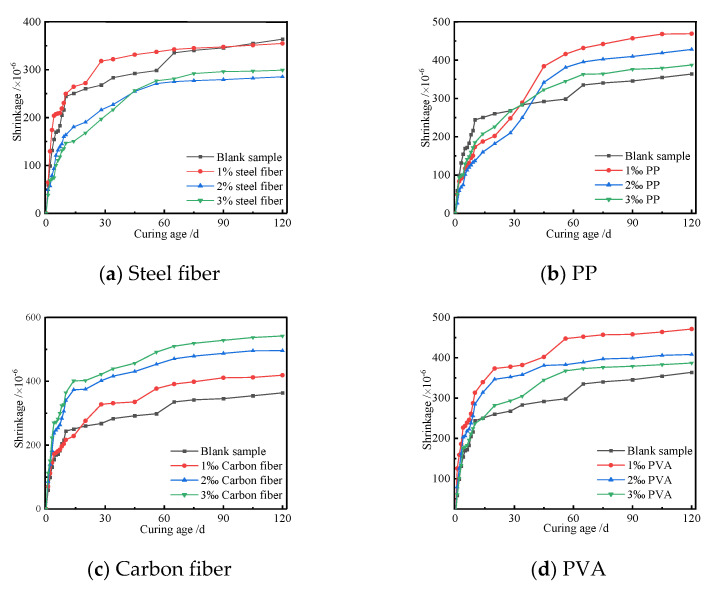
Shrinkage curves of the UHPC with different fibers and curing ages.

**Figure 12 polymers-14-01956-f012:**
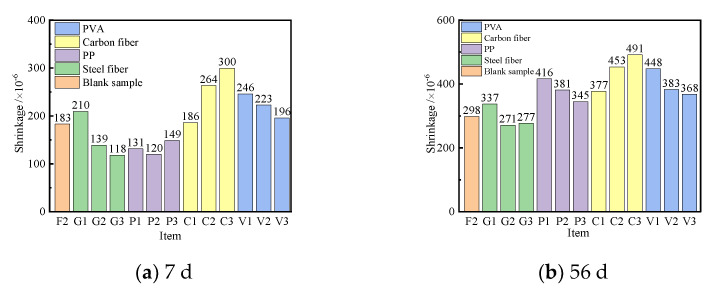
Shrinkage curves of the UHPC cured for 7 d and 56 d.

**Figure 13 polymers-14-01956-f013:**
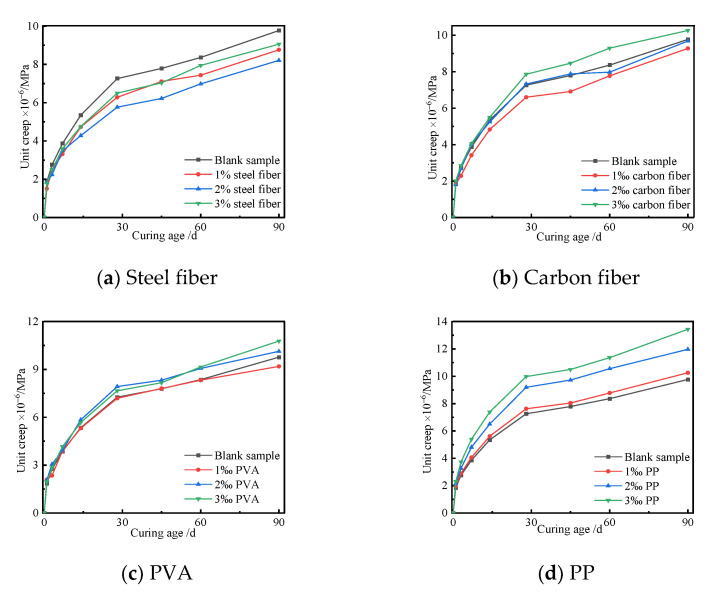
Unit creep curves of the UHPC with different types and contents of fibers.

**Figure 14 polymers-14-01956-f014:**
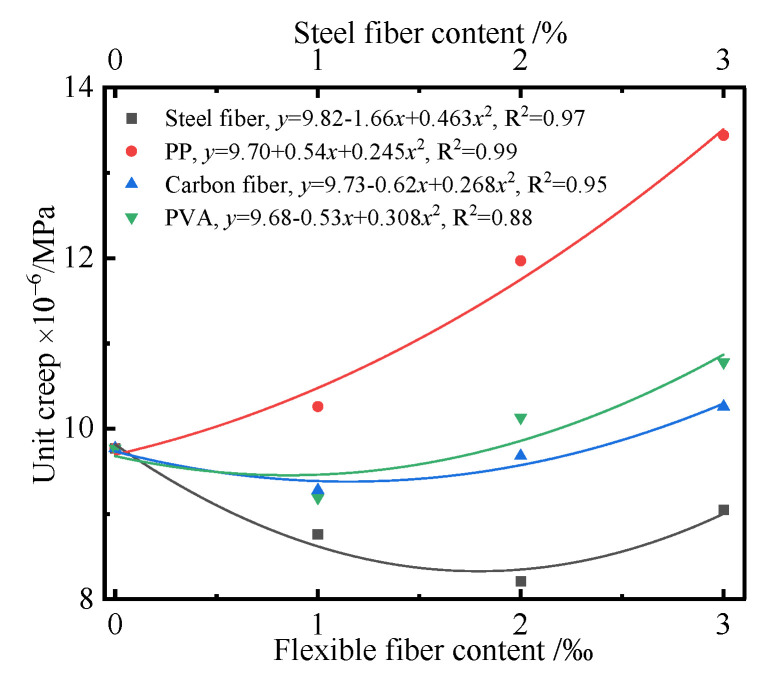
Unit creep curves of the UHPC cured for 90 d with the fibers content.

**Table 1 polymers-14-01956-t001:** Characteristic parameters of the fibers.

Items	Average Length/mm	Average Diameter of Monofilament/μm	Apparent Density/g·cm^−3^	Tensile Strength/MPa	Elasticity Modulus/GPa
Carbon fiber	6	6~14	0.4	4900	240
PVA	12	20~21	6~8	1400~1600	35~39
PP	12	18~48	>15	>358	>3.5
Steel fiber	32	2000	7.65	>1100	200~210

**Table 2 polymers-14-01956-t002:** The mix proportion design of UHPC.

Items	*W/C*	Total Cementitious Materials Weight /kg·m^−3^	Cement/kg·m^−3^	Fly Ash/kg·m^−3^	Silica/kg·m^−3^	I Type of Coarse Aggregate/kg·m^−3^	Ⅱ Type of Coarse Aggregate/kg·m^−3^	Sand/kg·m^−3^	Sand Ratio/%	Defoamer Dosage/%	Fiber Type	Fiber Content/%	Water Reducer/%	Compressive Strength at 28 d/MPa
S1	0.2	600	420	108	72	258.9	866.7	554.4	33				3.40	86.9
S2	0.2	600	420	108	72	243.4	815.0	621.6	37				2.80	89.8
S3/F2/H3	0.2	600	420	108	72	228.0	763.2	688.8	41				2.60	100.4
S4	0.2	600	420	108	72	212.5	711.5	756.0	45				3.40	85.4
F1	0.2	550	385	99	66	236.1	790.5	713.4	41				3.00	106
F3	0.2	650	455	117	78	219.8	736.0	664.2	41				2.20	104.5
F4	0.2	700	490	126	84	211.7	708.7	639.6	41				1.50	107.3
F5	0.2	750	525	135	90	203.6	681.5	615.0	41				0.90	110.5
H1	0.16	600	420	108	72	228.0	763.2	688.8	41				6.40	92.1
H2	0.18	600	420	108	72	228.0	763.2	688.8	41				5.30	91.2
H4	0.22	600	420	108	72	228.0	763.2	688.8	41				2.50	95.5
H5	0.24	600	420	108	72	228.0	763.2	688.8	41				1.50	96.1
X1	0.2	600	420	108	72	228.0	763.2	688.8	41	0.5			—	101.9
X2	0.2	600	420	108	72	228.0	763.2	688.8	41	1			—	105.8
X3	0.2	600	420	108	72	228.0	763.2	688.8	41	1.5			—	114
C1	0.2	600	420	108	72	228	763.2	688.8			Carbon fiber	0.1	3.90	108.2
C2	0.2	600	420	108	72	228	763.2	688.8				0.2	5.30	108.8
C3	0.2	600	420	108	72	228	763.2	688.8	41	1.5		0.3	5.80	105.5
P1	0.2	600	420	108	72	228	763.2	688.8			PP	0.1	5.70	89.4
P2	0.2	600	420	108	72	228	763.2	688.8				0.2	7.10	91.3
P3	0.2	600	420	108	72	228	763.2	688.8				0.3	7.90	83.4
V1	0.2	600	420	108	72	228	763.2	688.8			PVA	0.1	4.20	96.5
V2	0.2	600	420	108	72	228	763.2	688.8				0.2	4.10	99.5
V3	0.2	600	420	108	72	228	763.2	688.8				0.3	3.90	102.4
G1	0.2	600	420	108	72	228	763.2	688.8			Steel fiber	1	2.70	109
G2	0.2	600	420	108	72	228	763.2	688.8				2	3.80	122.3
G3	0.2	600	420	108	72	228	763.2	688.8				3	4.20	120.8

Noted: The volume content was used for steel fiber, and the mass content was used for carbon fiber, PP, and PVA. S represents the sand ratio, i.e., S1, S2, S3, and S4 are the sand ratios of 33%, 37%, 41%, and 45%, respectively. H is the water-to-cement ratio, F stands for the fly ash content, and X represents the defoamer dosage. Furthermore, C, P, V, and G are the contents of carbon fibers, PP, PVA, and steel fibers, respectively.

## Data Availability

The data that support the findings of this study are available from the corresponding author.

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
