# Peer review of "Effects of Type and Content of Fibers, Water-to-Cement Ratio, and Cementitious Materials on the Shrinkage and Creep of Ultra-High Performance Concrete"

_polymers, 2022, doi:10.3390/polym14101956_

Round 1
Reviewer 1 Report
In this paper, the effects of type and contents of fibers, water-to-cement ratio, and cementitious materials on shrinkage and creep of UHPC were investigated. The reviewer thought that the authors provided proper experiments and good results, showing the novel contribution in similar fields. Thus, I hope to accept this paper.
Author Response
Dear Reviewer,
Thank you very much for your careful and constructive comments on the manuscript (No. Polymers-1718374, title: Effects of type and content of fibers, water to cement ratio and cementitious materials on the shrinkage and creep of ultra-high performance concrete). Your comments are very helpful for improving our paper. According to your recommendations, we revised the manuscript and made correction which we hope meet with approval, and the modified parts were all marked in blue.
I hope you are satisfied with the revised version, however, if there is more question, we are willing to revise it again.
Yours sincerely,
Liu Peng
Next section is the reply to reviewers’ comments:
Reviewer #1:
In this paper, the effects of type and contents of fibers, water-to-cement ratio, and cementitious materials on shrinkage and creep of UHPC were investigated. The reviewer thought that the authors provided proper experiments and good results, showing the novel contribution in similar fields. Thus, I hope to accept this paper.
(Thanks for your utmost effort and full support. According to your comment, the authors modified some grammar and sentences, which were marked in blue.)
Please accept our sincere thankfulness for your utmost effort and full support. Thank you for reviewing our manuscript.

Reviewer 2 Report
In this paper the effects of type and content of fibers, water to cement ratio, content of cementitious materials on the shrinkage and creep of ultra-high performance concrete were investigated from an experimental point of view. The paper is well-structured, and the content is within the scope of the journal. Conclusions are sound and based on evidence provided by the results. Therefore, I can recommend the paper to be considered for publication. Authors are address the following queries:[1] In the introduction section I will recommend that the objectives of the study will be presented in a separate paragraph with clear definition.
[2] Can the experimental proposed approach be developed using advanced image correlation techniques, for instance, to measure over an entire surface the shrinkage (Eq. 1)? How is the current state-of-the-art regarding this aspect?
[3] Eq. (1): after the equation \noindent is required.
[4] line 212 p.6: Fig.2 can be rather written as "Figure~2".
[5] line 216 p.6: I believe there is no need to declare "..their relationship, as Eq.(3)...". It is explicit that you are referring to Eq. (3), so this can be removed on the text, throughout all the manuscript.
Author Response
Dear Reviewer,
Thank you very much for your careful and constructive comments on the manuscript (No. Polymers-1718374, title: Effects of type and content of fibers, water to cement ratio and cementitious materials on the shrinkage and creep of ultra-high performance concrete). Your comments are very helpful for improving our paper. According to your recommendations, we revised the manuscript and made correction which we hope meet with approval, and the modified parts were all marked in blue.
I hope you are satisfied with the revised version, however, if there is more question, we are willing to revise it again.
Yours sincerely,
Liu Peng
Next section is the reply to reviewers’ comments:
Reviewer #2:
In this paper the effects of type and content of fibers, water to cement ratio, content of cementitious materials on the shrinkage and creep of ultra-high performance concrete were investigated from an experimental point of view. The paper is well-structured, and the content is within the scope of the journal. Conclusions are sound and based on evidence provided by the results. Therefore, I can recommend the paper to be considered for publication. Authors are address the following queries:
[1] In the introduction section I will recommend that the objectives of the study will be presented in a separate paragraph with clear definition.
(OK. According to your comment, we modified the corresponding part, which were marked in blue.)
[2] Can the experimental proposed approach be developed using advanced image correlation techniques, for instance, to measure over an entire surface the shrinkage (Eq. 1)? How is the current state-of-the-art regarding this aspect?
(OK. As you say, the advanced image correlation techniques can be used to determine the shrinkage of specimens. However, the shrinkage of concrete specimen was performed by distance-marking method, which was recommended by Chinese standard of GB/T 50082-2009. This method was regarded as a prioritized approach to determine the shrinkage of concrete specimen in China, so we conducted this testing and calculated the shrinkage by Eq.(1).)
[3] Eq. (1): after the equation \noindent is required.
(OK. According to your comment, we modified the format problem, which were marked in blue.)
[4] line 212 p.6: Fig.2 can be rather written as "Figure~2".
(OK. According to your comment, we modified the corresponding parts, which were marked in blue.)
[5] line 216 p.6: I believe there is no need to declare "..their relationship, as Eq.(3)...". It is explicit that you are referring to Eq. (3), so this can be removed on the text, throughout all the manuscript.
(OK. According to your comment, we modified the corresponding part, which was marked in blue.)
Please accept our sincere thankfulness for your utmost effort and full support. Thank you for reviewing our manuscript.
